# ReactEmbed: A Plug-and-Play Module for Unifying Protein-Molecule Representations Guided by Biochemical Reaction Networks

## Abstract

State-of-the-art models represent proteins and molecules in separate embedding manifolds, limiting the modeling of systemic biological processes. We introduce ReactEmbed, a lightweight, plug-and-play module that bridges this gap. ReactEmbed leverages biochemical reaction networks as a source of functional context, based on the principle that co-participation in reactions defines a shared functional scope. The module aligns frozen embeddings from models like ESM-3 and MolFormer into a unified space using a weighted reaction graph and a specialized sampling strategy. This process enriches unimodal embeddings and enables strong performance on cross-domain benchmarks. ReactEmbed offers a practical method to unify biological representations without costly retraining. The code and database are available for open use[1].

## 1 Introduction

The fields of computational biology and drug discovery have been revolutionized by large-scale foundation models. For proteins, models like ESM-3 Hayes et al. (2025) and ProtBERT Brandes et al. (2021) have learned deep representations from sequence data, while for molecules, models like MolFormer Ross et al. (2022) have done the same from SMILES strings. While immensely powerful, these models operate in separate computational universes. Due to their fundamentally different data modalities (e.g., amino acid sequences vs. SMILES strings) and specialized architectures, their representations are incompatible, creating a fundamental bottleneck. Biology is driven by the systemic interactions between entities, yet our best models lack a unified language to describe the functional relationships between proteins and molecules. Prior attempts to bridge this gap have focused on narrow aspects of interaction, primarily physical complementarity. Interaction-centric models like DrugCLIP Gao et al. (2023) excel at predicting if a molecular "key" fits into a protein's "lock" (the binding pocket). This makes them highly effective for virtual screening, but their representations are specialized for this single task and may not generalize to broader biological functions. Similarly, structure-centric models like Uni-Mol Zhou et al. (2023) unify entities through their 3D geometry. This is a powerful approach for tasks like pose prediction but can miss functional equivalences between structurally dissimilar entities. The common limitation is a focus on how entities physically fit together, rather than what they functionally achieve together. In this work, we introduce a new paradigm for unifying biological representations based on functional semantics.

Our key invention is the insight that biochemical reaction networks serve as a vast, curated semantic nexus for biological function. Co-participation in a reaction is an explicit, unambiguous signal of a functional role, regardless of structural similarity or direct binding. We operationalize this paradigm in ReactEmbed, a lightweight, computationally efficient plug-and-play enhancement module. ReactEmbed takes any off-the-shelf, frozen embeddings and aligns them into a unified, functionally-aware space. It does this through a novel relational learning architecture that interprets a weighted reaction graph, systematically learning the functional context of each entity. This approach yields a cascade of demonstrated benefits. First, the alignment process is mutually beneficial, providing context that enriches each unimodal representation for its own domain-specific tasks. Second, this enrichment leads to strong performance across a wide array of cross-domain benchmarks, from drug-target interaction to binding affinity prediction.

---

[1]https://anonymous.4open.science/r/ReactEmbeded-A283/README.md

The main contributions of this work are: (1) A new paradigm for joint representation learning that leverages systemic biochemical reaction networks to capture functional semantics. (2) ReactEmbed, a lightweight, plug-and-play module that enhances and unifies existing, frozen embeddings without costly retraining or fine-tuning. (3) Demonstration that this approach improves cross-domain task performance and enriches unimodal representations, validated across a diverse range of benchmarks. To facilitate broader adoption, the ReactEmbed code and database are https://anonymous.4open.science/r/ReactEmbeded-A283/README.md.

## 2 Related Work

Our work builds on foundations in unimodal representation learning and introduces a new paradigm for joint protein-molecule representations. The following sections provide an exhaustive review of the intellectual landscape, charting the parallel evolution of protein and molecule modeling before critically examining existing joint representation paradigms. This contextualization highlights the specific limitations of current approaches and motivates the development of ReactEmbed, a novel framework grounded in functional semantics derived from biochemical reaction networks.

### 2.1 Representation Learning in Proteins

Sequence-based protein language models (PLMs), primarily transformer-based, have seen remarkable success Brandes et al. (2021); Hayes et al. (2025). These models leverage vast unlabeled sequence repositories, applying self-supervised objectives like Masked Language Modeling (MLM) to learn representations that implicitly capture co-evolution, structure, and function Rao et al. (2019); Brandes et al. (2021). The ESM family Hayes et al. (2025) and models like ProtBERT Brandes et al. (2021) have become powerful feature extractors for diverse downstream tasks. Following breakthroughs in structure prediction Jumper et al. (2021), structure-based methods have also gained prominence. These models, often Graph Neural Networks (GNNs), directly encode 3D geometry Zhang et al. (2023). GearNet, for example, uses a relational GNN pre-trained on the AlphaFold Database with geometric self-supervised tasks Zhang et al. (2023).

### 2.2 Representation Learning in Molecules

Molecular representation has followed a similar trajectory. Sequence-based models like ChemBERTa Chithrananda et al. (2020a) and MolFormer Ross et al. (2022) apply transformers to SMILES strings, treating chemistry as a "language" Ross et al. (2022). In parallel, GNN-based methods naturally capture molecular topology (atoms as nodes, bonds as edges). Models like Message Passing Neural Networks (MPNNs) Gilmer et al. (2017) and contrastive learning frameworks like MolCLR Wang et al. (2022b) have proven highly effective by learning from the graph structure itself.

### 2.3 Joint Protein-Molecule Representations

Joint representation learning aims to place proteins and molecules in a unified space. **Interaction-centric models** (e.g., DrugCLIP Gao et al. (2023)) use contrastive architectures specialized for virtual screening but may lack broader functional context. **Structure-centric models** (e.g., Uni-Mol Zhou et al. (2023)) unify entities via 3D geometry, effective for pose prediction but potentially missing functional equivalences between structurally dissimilar entities.

**Graph and KG Methods.** Methods like KG-DTI, KGNN Wang et al. (2022a); Hu et al. (2022), and recent architectures like DCGAT-DTIAbir et al. (2026) explicitly model relational data. However, these are often *transductive* or graph-dependent at inference, requiring known graph neighbors to generate predictions. **ReactEmbed** differs by being fully *inductive*. It functions as a modular enhancement for frozen Foundation Models, utilizing the reaction graph only during training to imprint context. At inference time, it operates solely on entity features, enabling predictions for "cold-start" entities with no known reaction history.

# 3 The ReactEmbed Framework

ReactEmbed is a novel method designed to enhance and align protein and molecule representations by integrating biochemical reaction data with pre-trained embeddings, as illustrated in Figure 1.

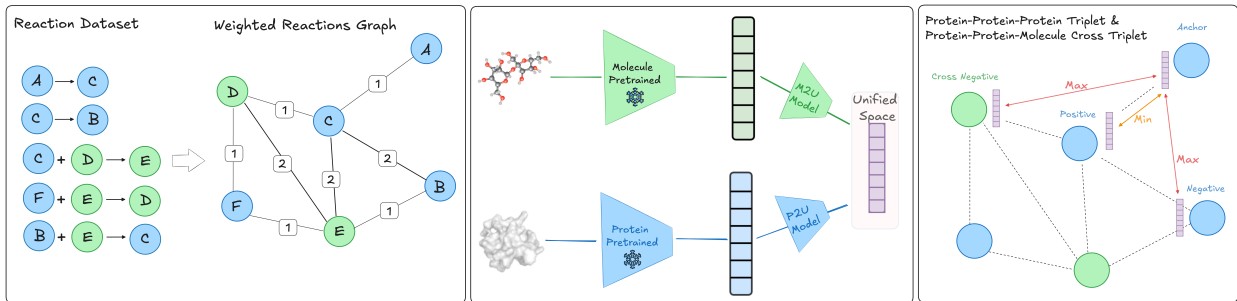

Figure 1: Overview of the ReactEmbed framework. **Left:** A toy reaction dataset is converted into a weighted reaction graph. We move beyond simple co-occurrence counts, using normalized association scores (PPMI) as edge weights to capture functional specificity and dampen the effect of common 'hub' nodes. **Middle:** The ReactEmbed model architecture. Frozen, domain-specific pre-trained embeddings ($E_{pre}^p, E_{pre}^m$) are projected into a unified space via lightweight, trainable P2U (Protein to Unified) and M2U (Molecule to Unified) MLP modules. **Right:** Illustration of our advanced sampling strategy. An anchor (blue circle) samples a functionally-specific positive partner (green circle) from its 1-hop neighborhood, weighted by PPMI. It also samples challenging 'hard negatives' (red circles) from its $k$-hop neighborhood ($k \in \{2, 3, 4, 5\}$) for both intra-domain (e.g., another protein) and cross-domain (e.g., a molecule) alignment, forcing the model to learn fine-grained distinctions.

## 3.1 Methodology

### 3.1.1 Method Overview and Objectives

Given sets of proteins $\mathcal{P}$ and molecules $\mathcal{M}$, along with a database of reactions $\mathcal{R}$, ReactEmbed enhances individual protein and molecule representations. First, we construct a comprehensive weighted reaction graph, where edge weights are derived from normalized association scores (PPMI) to capture functional specificity. This graph represents protein-protein, protein-molecule, and molecule-molecule associations. Second, we leverage this graph through a novel relational contrastive learning framework, which employs an advanced sampling strategy (hard negative mining) to efficiently preserve domain-specific information while learning cross-domain functional relationships. Algorithm 1 summarizes our methodology.

### 3.1.2 Reaction-to-Graph Conversion and PPMI Weighting

The first step is to convert the reaction dataset $\mathcal{R}$ into a weighted, undirected graph $G = (V, E, W_{PPMI})$. The set of vertices $V = \mathcal{P} \cup \mathcal{M}$ includes all proteins and molecules. A simple co-occurrence count $C(e_i, e_j)$ is a problematic edge weight, as it is dominated by ubiquitous "hub" nodes (e.g., ATP, water) that are functionally general. To capture functional specificity, we compute edge weights using Positive Pointwise Mutual Information (PPMI)Church & Hanks (1990), a standard measure of association from information theory, to mitigate this "hub problem." First, we compute the raw co-occurrence counts $C(e_i, e_j)$ as shown in Algorithm 1 (lines 2-7). From these, we estimate the probabilities $P(e_i, e_j)$ and the marginal probabilities $P(e_i)$ and $P(e_j)$. The PMI is then:

$$PMI(e_i, e_j) = \log\left(\frac{P(e_i, e_j)}{P(e_i)P(e_j)}\right) \qquad (1)$$

PMI measures how much more likely two entities are to co-occur than if they were independent. We use the Positive PMI, $W_{PPMI}(e_i, e_j) = \max(0, PMI(e_i, e_j))$, as the final edge weight. This hub-dampened weight is high for pairs that are specifically associated and low for pairs that co-occur merely by chance or due to one entity being a hub.

---

**Algorithm 1** Advanced Relational Learning for Embedding Enhancement

---

**Require:** Reaction dataset $\mathcal{R}$; Protein set $\mathcal{P}$, Molecule set $\mathcal{M}$.
**Require:** Pre-trained, frozen embedders $E_{pre}^p, E_{pre}^m$.
    Let $h(x)$ denote the embedding enhancement module.
    **Phase 1: Graph Construction and PPMI Weighting**
1: Initialize $G(V, E)$ with $V = \mathcal{P} \cup \mathcal{M}$.
2: Initialize co-occurrence count matrix $C = \mathbf{0}$.
3: **for** $r \in \mathcal{R}$ **do**
4:     Let $E_r \subseteq V$ be the set of entities in reaction $r$.
5:     **for** each distinct pair $\{e_i, e_j\} \subseteq E_r$ **do**
6:         $C(e_i, e_j) \leftarrow C(e_i, e_j) + 1$.
7:     **end for**
8: **end for**
9: Let $N_{total} = \sum_{i,j} C(e_i, e_j)$ be total co-occurrences.
10: Let $D(e_i) = \sum_j C(e_i, j)$ be marginal occurrences of $e_i$.
11: Initialize weight matrix $W_{PPMI} = \mathbf{0}$.
12: **for** each pair $\{e_i, e_j\}$ where $C(e_i, e_j) > 0$ **do**
13:     $P(e_i, e_j) \leftarrow C(e_i, e_j)/N_{total}$
14:     $P(e_i) \leftarrow D(e_i)/N_{total}; P(e_j) \leftarrow D(e_j)/N_{total}$
15:     $PMI(e_i, e_j) \leftarrow \log\left(\frac{P(e_i,e_j)}{P(e_i)P(e_j)}\right)$
16:     $W_{PPMI}(e_i, e_j) \leftarrow \max(0, PMI(e_i, e_j))$
17: **end for**
    **Phase 2: Relational Learning via Advanced Sampling**

18: **while** not converged **do**
19:     Sample anchor $x_a \sim \text{Uniform}(V)$.
20:     Let $N(x_a)$ be the set of 1-hop neighbors of $x_a$.
21:     **Hub-Dampened Positive Sampling:**
22:     Sample $x_p$ from $N(x_a)$ with probability $P(x_p|x_a) \propto W_{PPMI}(x_a, x_p)$.
23:     **Graph-Based Hard Negative Sampling:**
24:     Let $D_a$ be domain of $x_a$; $D_o$ be the opposite domain.
25:     Sample $k \sim \text{Uniform}(\{2, 3, 4, 5\})$.
26:     Let $N_k(x_a)$ be the set of $k$-hop neighbors of $x_a$.
27:     $N_{k,intra} \leftarrow (N_k(x_a) \cap D_a) \setminus N(x_a)$
28:     $N_{k,cross} \leftarrow (N_k(x_a) \cap D_o) \setminus N(x_a)$
29:     Sample $x_{n,intra} \sim \text{Uniform}(N_{k,intra}$ or $D_a$ if $N_{k,intra} = \emptyset)$.
30:     Sample $x_{n,cross} \sim \text{Uniform}(N_{k,cross}$ or $D_o$ if $N_{k,cross} = \emptyset)$.
31:     $\mathcal{L}_{intra} \leftarrow \max(0, m + d(h(x_a), h(x_p)) - d(h(x_a), h(x_{n,intra})))$.
32:     $\mathcal{L}_{cross} \leftarrow \max(0, m + d(h(a), h(p)) - d(h(a), h(x_{n,cross})))$.
33:     $\mathcal{L}_{total} \leftarrow \alpha \mathcal{L}_{intra} + (1 - \alpha)\mathcal{L}_{cross}$.
34:     Update parameters of $h(x)$ using $\nabla \mathcal{L}_{total}$.
35: **end while**
36: **return** Trained enhancement module parameters $\theta_{P2U}, \theta_{M2U}$.

---

### 3.1.3 Relational Learning with Advanced Sampling

Our framework aligns the disparate embedding manifolds of proteins and molecules into a single, functionally coherent space. A key design choice of our "plug-and-play" module is to use frozen pre-trained embedding functions:

$$E_{pre}^p : \mathcal{P} \to \mathbb{R}^{d_p}, \qquad E_{pre}^m : \mathcal{M} \to \mathbb{R}^{d_m}$$

This is a deliberate strategy to maintain the rich, general features learned by these large models and to ensure our enhancement module is computationally efficient. These frozen embeddings are projected into a unified latent space of dimension $d_s$ using our trainable enhancement module. This module consists of two lightweight multi-layer perceptrons (MLPs), P2U (Protein to Unified) and M2U (Molecule to Unified). These are two-layer MLPs with ReLU activation, chosen to keep the module computationally efficient and avoid overfitting. The final unified embedding for

any entity $x$ is given by:

$$E_{\text{unified}}(x) = \begin{cases} \text{P2U}(E_{\text{pre}}^p(x); \theta_{P2U}), & \text{if } x \in \mathcal{P} \\ \text{M2U}(E_{\text{pre}}^m(x); \theta_{M2U}), & \text{if } x \in \mathcal{M} \end{cases} \tag{2}$$

The module is trained through a specialized relational learning process that leverages the weighted reaction graph. The core of this process is an advanced sampling strategy:

- **Hub-Dampened Positive Sampling:** For a given anchor $x_a$, we sample a positive partner $x_p$ from its 1-hop neighborhood $N(x_a)$ with a probability proportional to the PPMI edge weight, $P(x_p|x_a) \propto W_{PPMI}(x_a, x_p)$. This ensures the training signal comes from functionally specific, not just common, associations.

- **Graph-Based Hard Negative Sampling:** Random negative sampling is inefficient. We instead employ a hard negative mining strategy based on graph topology. We sample negatives from the $k$-hop neighborhood of the anchor (where $k$ is randomly chosen from $\{2, 3, 4, 5\}$), which are entities that are functionally related but not direct partners. This forces the model to learn fine-grained distinctions.

- **Intra- and Cross-Domain Objectives:** This hard sampling is applied to our two core objectives: (1) *Intra-Domain Preservation*, which samples a hard negative $x_{n,intra}$ from the same domain (e.g., a $k$-hop protein, excluding 1-hop neighbors) to preserve fine-grained structure, and (2) *Cross-Domain Alignment*, which samples a hard negative $x_{n,cross}$ from the opposite domain (e.g., a $k$-hop molecule, excluding 1-hop neighbors) to drive alignment.

These two objectives are balanced in a multi-faceted loss function. The total loss is:

$$\mathcal{L}_{\text{total}} = \alpha \mathcal{L}_{\text{intra}} + (1 - \alpha) \mathcal{L}_{\text{cross}} \tag{3}$$

Here, the $\mathcal{L}_{\text{intra}}$ term enforces structural preservation within each domain, while the $\mathcal{L}_{\text{cross}}$ term drives the functional alignment between them. Each component is a margin-based triplet loss:

$$\begin{aligned} \mathcal{L}_{\text{intra}} &= \max(0, m + d(h_a, h_p) - d(h_a, h_{n,intra})) \\ \mathcal{L}_{\text{cross}} &= \max(0, m + d(h_a, h_p) - d(h_a, h_{n,cross})) \end{aligned} \tag{4}$$

where $h_x = E_{\text{unified}}(x)$ represents the projected embedding of an entity $x$, $d(\cdot, \cdot)$ is the Cosine distance (chosen to focus the model on the angular relationship between embeddings rather than their magnitude), and $m$ is the margin. This advanced framework ensures that specific functional signals are learned efficiently by focusing the model on challenging examples.

## 4 Empirical Evaluation

### 4.1 Tasks

The empirical evaluation utilizes tasks from four primary areas: molecular properties, protein characteristics, protein-protein interactions, and molecule-protein interactions. This set of tasks aligns with those commonly employed in previous works to benchmark models within these respective domains Hayes et al. (2025); Brandes et al. (2021); Wang et al. (2022b); Ross et al. (2022); Zhang et al. (2023); Wang et al. (2022b); Chithrananda et al. (2020b); Rao et al. (2019); Xu et al. (2022). For molecule-protein interaction prediction, we evaluated DrugBank Wishart et al. (2018) focused on drug-target interaction prediction, and BindingDB Liu et al. (2007), which provides binding affinity measurements. For protein-protein interaction prediction, we utilized three complementary datasets: HumanPPI Pan et al. (2010) and YeastPPI Guo et al. (2008), which evaluate interaction prediction capabilities across human and yeast organisms, respectively; and PPIAffinity Moal & Fernández-Recio (2012), which provides quantitative measurements of binding strength between protein pairs. For protein property prediction, we evaluated our model on three distinct datasets: BetaLactamase Gray et al. (2018), which measures activity values of TEM-1 beta-lactamase protein first-order mutants; Stability Rocklin et al. (2017), which quantifies protein stability; and Cellular Component (GeneOntology)Consortium (2019), which focuses on cellular component classification. For molecular property

prediction, we evaluated three key datasets: BBBP Martins et al. (2012), which measures blood-brain barrier penetration; FreeSolv Mobley & Guthrie (2014), which examines hydration free energy; and CEP Lopez et al. (2016), which estimates photovoltaic efficiency.

## 4.2 Baseline Methods

We selected state-of-the-art pre-trained models to serve as the base embeddings for our "plug-and-play" module. Our selection criteria prioritized models that represent the distinct prevailing paradigms in the field:

- **Sequence & Structure Baselines:** For proteins, we used ESM-3 Hayes et al. (2025) and GearNet Zhang et al. (2023). For molecules, we chose MolFormer Ross et al. (2022) and MolCLR Wang et al. (2022b).

- **Multimodal Comparisons:** We compare against **DrugCLIP** Gao et al. (2023) and **Uni-Mol** Zhou et al. (2023) as canonical representatives of the *Interaction-Centric* and *Structure-Centric* paradigms. To ensure our evaluation captures the latest advancements, we also compare against two very recent graph-based architectures: **DCGAT-DTI** Abir et al. (2026), which utilizes dynamic cross-graph attention, and **MRHormer** Zhang et al. (2026), a multi-scale heterogeneous graph transformer.

## 4.3 Experimental Setup

Our study utilized publicly available datasets and their standard splits to ensure fair comparison. All data loading and splitting were implemented via the TorchDrug library Zhu et al. (2022). For datasets with canonical pre-defined splits, we adopted them directly. For datasets without pre-defined splits (e.g., molecular properties), we utilized deterministic Ordered Scaffold Splitting (80% train, 10% valid, 10% test) to ensure a rigorous evaluation of structural generalization out-of-distribution. For downstream interaction tasks, we strictly utilized the pre-defined benchmark partitions without modification. The reaction data for training ReactEmbed was sourced from the Reactome database, a high-quality, manually curated database.

The resulting reaction graph contained 8,692 protein and 2,204 molecule nodes. The ReactEmbed projection layers (P2U and M2U) are two-layer MLPs (projecting from the base model dimension, e.g., 768, to a hidden dimension of 1024, and back to a final unified embedding space of 768) with ReLU activation. Key hyperparameters were tuned via grid search on a held-out validation set of reactions, optimizing the contrastive loss. We searched learning rates over $\{1e-5, 5e-5, 1e-4\}$ and the triplet margin $m$ over $\{0.1, 0.3, 0.5\}$. The final parameters used for all experiments were a learning rate of $5e-5$, a margin of $0.1$, and a loss balancing weight $\alpha$ (Equation 3) of $0.5$. The distance function $d(\cdot, \cdot)$ used was Cosine distance. Training was performed for 50 epochs with a batch size of 256 using the AdamW optimizer.

Following standard evaluation protocol Rao et al. (2019), we assess representation quality by training a single linear layer on top of the frozen embeddings (either the original "Baseline" or our "ReactEmbed" enhanced embeddings) for each downstream task. To ensure a strict and fair comparison, all baseline models were initialized using their official, validated checkpoints. The results reported in Tables 1 and 2 were then reproduced completely within our framework by extracting frozen embeddings and training a linear probe using the exact same data splits and evaluation protocol as ReactEmbed, rather than being cited directly from prior literature. To ensure robustness, all experiments were repeated 10 times with different random seeds, and we report the mean and standard deviation of the results.

To maintain consistency with prior work, we adopt their standard evaluation metrics: AUC for classification and Root Mean Square Error (RMSE) for regression. Statistical significance in Tables 1 and 2 was determined using an independent two-sample t-test ($p < 0.05$). The computational complexity of the graph construction (Phase 1) is proportional to the number of reactions and the square of the number of participants in each reaction, followed by a pass over all co-occurrence pairs to compute PPMI. The relational learning phase (Phase 2) scales with the number of training epochs and the number of edges sampled. For the experiments presented in this work, constructing the full reaction graph from the Reactome dataset took approximately 45 minutes on a standard CPU. The complete training of the ReactEmbed module for 50 epochs required 15 minutes on a single NVIDIA L40 GPU.

### 4.4 Data Leakage Considerations

A critical aspect of our methodology is the prevention of data leakage between the Reactome training graph and the downstream interaction test sets (BindingDB, DrugBank, and PPI). Crucially, we utilized the standard, unmodified test sets provided by the TorchDrug benchmarks for all interaction tasks. To prevent leakage while maintaining comparability with these standard benchmarks, our sanitization was applied exclusively to the pre-training graph. To ensure a rigorous evaluation of the model's predictive power on novel interactions, we implemented a stringent data splitting protocol. Rather than removing the entities themselves (which would deplete the graph of valuable contextual information), we explicitly identified and removed any direct edge from the Reactome graph that represented a co-occurrence between two entities found to be homologous or analogous to a known pair in one of the downstream test sets.

Further analysis of different leakage prevention strategies is provided in Appendix A.

## 5 Results and Analysis

### 5.1 Performance on Downstream Tasks

As shown in Tables 2 and 1, applying the ReactEmbed module yields broad performance improvements across the majority of the 11 downstream tasks. The most substantial gains are observed in tasks centered on molecular interactions, where capturing functional context is critical. While ReactEmbed improved performance in the vast majority of settings, we observed minor degradation in a few configurations (e.g., GearNet on YeastPPI). This may suggest that for certain models or tasks, the functional semantics from reaction data may not perfectly align with the existing features, a phenomenon we plan to investigate in future work.

#### 5.1.1 Substantial Gains on Interaction-Centric Tasks

The most substantial gains are observed in tasks centered on molecular interactions, where capturing functional context is critical. For example, on the DrugBank drug-target prediction task, ReactEmbed improves performance by up to 10.28%. Similarly, on the BindingDB affinity prediction task, it reduces error by as much as 13.79%. This demonstrates that our method effectively captures the complex functional relationships that govern protein-molecule binding (while acknowledging the data leakage considerations discussed in Section 4.4).

#### 5.1.2 A Generally Applicable Enhancement for Sequence and Structure Models

This functional context generally enhances both sequence-based (ESM3, ProtBert) and structure-based (GearNet) models. This indicates that reaction data provides a complementary signal that enriches existing representations, regardless of their architectural foundation. For instance, the structure-aware GearNet sees a 15.6% error reduction on protein stability, while the sequence-based ESM3 improves by 13.6% on YeastPPI.

#### 5.1.3 Functional Context Appears Most Crucial for Complex Tasks

ReactEmbed's impact is most pronounced on more challenging tasks with lower baseline scores, such as YeastPPI and protein stability. This suggests our approach offers a crucial source of information for complex cases where sequence or structure alone are insufficient to capture the full picture. These results validate that ReactEmbed is a robust and practical method for enhancing and unifying biological representations.

### 5.2 Comparison with Multimodal Baselines

To further situate our work, we compare ReactEmbed against state-of-the-art multimodal models on the two primary cross-domain tasks: DrugBank (interaction classification) and BindingDB (affinity regression). As shown in Table 3, existing models are highly specialized, reflecting their training objectives.

The interaction-centric DrugCLIP, which is pre-trained to distinguish true from false interactions, excels at the Drug-Bank classification task. However, its representation is not optimized for the fine-grained nuances of affinity prediction, leading to weaker performance on BindingDB. Conversely, the structure-centric Uni-Mol, which is designed to predict

Table 1: Evaluation of ReactEmbed for regression tasks (RMSE, lower is better). Results are reported as Mean $\pm$ 95% Confidence Interval (over 10 runs). "Baseline" refers to the original frozen embeddings. Baseline results were reproduced internally using the official validated model checkpoints and our unified linear probe evaluation framework.

| Protein Model | Molecular Model | ReactEmbed (95% CI) | Baseline (95% CI) |
|---|---|---|---|
| | | FreeSolv | |
| ESM3 | MolCLR | $4.29 \pm 0.09$ | $4.26 \pm 0.09$ |
| ESM3 | MolFormer | $\mathbf{2.85 \pm 0.06}$ | $3.12 \pm 0.07$ |
| GearNet | MolCLR | $4.21 \pm 0.10$ | $4.26 \pm 0.09$ |
| GearNet | MolFormer | $2.91 \pm 0.06$ | $3.12 \pm 0.07$ |
| ProtBERT | MolCLR | $4.21 \pm 0.09$ | $4.26 \pm 0.09$ |
| ProtBERT | MolFormer | $2.92 \pm 0.07$ | $3.12 \pm 0.07$ |
| | | CEP | |
| ESM3 | MolCLR | $1.99 \pm 0.04$ | $2.02 \pm 0.05$ |
| ESM3 | MolFormer | $1.64 \pm 0.03$ | $1.68 \pm 0.04$ |
| GearNet | MolCLR | $2.01 \pm 0.05$ | $2.02 \pm 0.05$ |
| GearNet | MolFormer | $1.63 \pm 0.04$ | $1.68 \pm 0.04$ |
| ProtBERT | MolCLR | $1.99 \pm 0.06$ | $2.02 \pm 0.05$ |
| ProtBERT | MolFormer | $\mathbf{1.62 \pm 0.04}$ | $1.68 \pm 0.04$ |
| | | BetaLactamase | |
| ESM3 | MolCLR | $\mathbf{0.26 \pm 0.01}$ | $0.32 \pm 0.02$ |
| ESM3 | MolFormer | $\mathbf{0.26 \pm 0.02}$ | $0.32 \pm 0.02$ |
| ProtBERT | MolCLR | $0.29 \pm 0.01$ | $0.32 \pm 0.02$ |
| ProtBERT | MolFormer | $0.31 \pm 0.01$ | $0.32 \pm 0.02$ |
| | | Stability | |
| ESM3 | MolCLR | $0.44 \pm 0.02$ | $0.45 \pm 0.04$ |
| ESM3 | MolFormer | $\mathbf{0.43 \pm 0.02}$ | $0.45 \pm 0.04$ |
| GearNet | MolCLR | $0.54 \pm 0.02$ | $0.64 \pm 0.02$ |
| GearNet | MolFormer | $0.55 \pm 0.03$ | $0.64 \pm 0.02$ |
| ProtBERT | MolCLR | $0.53 \pm 0.03$ | $0.53 \pm 0.02$ |
| ProtBERT | MolFormer | $0.51 \pm 0.02$ | $0.53 \pm 0.02$ |
| | | BindingDB | |
| ESM3 | MolCLR | $1.28 \pm 0.02$ | $1.48 \pm 0.03$ |
| ESM3 | MolFormer | $1.21 \pm 0.03$ | $1.40 \pm 0.02$ |
| GearNet | MolCLR | $1.28 \pm 0.03$ | $1.45 \pm 0.04$ |
| GearNet | MolFormer | $1.20 \pm 0.04$ | $1.36 \pm 0.03$ |
| ProtBERT | MolCLR | $1.24 \pm 0.02$ | $1.41 \pm 0.03$ |
| ProtBERT | MolFormer | $\mathbf{1.17 \pm 0.03}$ | $1.36 \pm 0.03$ |
| | | PPIAffinity | |
| ESM3 | MolCLR | $3.03 \pm 0.07$ | $3.32 \pm 0.07$ |
| ESM3 | MolFormer | $\mathbf{3.02 \pm 0.06}$ | $3.32 \pm 0.07$ |
| GearNet | MolCLR | $3.09 \pm 0.09$ | $3.69 \pm 0.09$ |
| GearNet | MolFormer | $3.10 \pm 0.08$ | $3.69 \pm 0.09$ |
| ProtBERT | MolCLR | $3.10 \pm 0.07$ | $3.32 \pm 0.12$ |
| ProtBERT | MolFormer | $3.14 \pm 0.07$ | $3.32 \pm 0.12$ |

3D poses and geometry, performs well on the BindingDB affinity task but is less effective for the broader functional classification task in DrugBank.

ReactEmbed is the only model to achieve state-of-the-art performance on *both* tasks. This demonstrates the generality of our approach. By learning functional semantics from reaction networks rather than specializing in physical interaction or 3D structure, ReactEmbed creates a unified representation that is broadly applicable to diverse protein-molecule challenges.

Table 2: Evaluation of ReactEmbed for classification tasks (AUC, higher is better). Results are reported as Mean $\pm$ 95% Confidence Interval (over 10 runs).Baseline results were reproduced internally using the official validated model checkpoints and our unified linear probe evaluation framework.

| Protein Model | Molecular Model | ReactEmbed (95% CI) | Baseline (95% CI) |
|---|---|---|---|
| **BBBP** | | | |
| ESM3 | MolCLR | $61.68 \pm 0.56$ | $58.23 \pm 0.59$ |
| ESM3 | MolFormer | $65.13 \pm 0.55$ | $64.92 \pm 0.56$ |
| GearNet | MolCLR | $59.36 \pm 0.61$ | $58.23 \pm 0.59$ |
| GearNet | MolFormer | $64.24 \pm 0.58$ | $64.92 \pm 0.56$ |
| ProtBERT | MolCLR | $64.86 \pm 0.55$ | $58.23 \pm 0.59$ |
| ProtBERT | MolFormer | $\mathbf{65.22 \pm 0.56}$ | $64.92 \pm 0.56$ |
| **GO-CC** | | | |
| ESM3 | MolCLR | $82.14 \pm 0.37$ | $81.00 \pm 0.40$ |
| ESM3 | MolFormer | $\mathbf{82.32 \pm 0.36}$ | $81.00 \pm 0.40$ |
| GearNet | MolCLR | $69.98 \pm 0.51$ | $69.63 \pm 0.53$ |
| GearNet | MolFormer | $69.97 \pm 0.52$ | $69.63 \pm 0.53$ |
| ProtBERT | MolCLR | $80.11 \pm 0.45$ | $79.58 \pm 0.46$ |
| ProtBERT | MolFormer | $80.18 \pm 0.44$ | $79.58 \pm 0.46$ |
| **DrugBank** | | | |
| ESM3 | MolCLR | $80.10 \pm 0.65$ | $76.34 \pm 0.71$ |
| ESM3 | MolFormer | $84.30 \pm 0.59$ | $76.70 \pm 0.68$ |
| GearNet | MolCLR | $78.32 \pm 0.71$ | $72.18 \pm 0.81$ |
| GearNet | MolFormer | $82.41 \pm 0.65$ | $74.73 \pm 0.78$ |
| ProtBERT | MolCLR | $83.52 \pm 0.61$ | $76.55 \pm 0.69$ |
| ProtBERT | MolFormer | $\mathbf{85.53 \pm 0.56}$ | $78.93 \pm 0.62$ |
| **HumanPPI** | | | |
| ESM3 | MolCLR | $93.90 \pm 0.17$ | $93.78 \pm 0.19$ |
| ESM3 | MolFormer | $\mathbf{94.88 \pm 0.15}$ | $93.78 \pm 0.19$ |
| GearNet | MolCLR | $86.28 \pm 0.46$ | $83.74 \pm 0.50$ |
| GearNet | MolFormer | $85.30 \pm 0.48$ | $83.74 \pm 0.50$ |
| ProtBERT | MolCLR | $91.52 \pm 0.31$ | $88.89 \pm 0.37$ |
| ProtBERT | MolFormer | $90.16 \pm 0.34$ | $91.52 \pm 0.31$ |
| **YeastPPI** | | | |
| ESM3 | MolCLR | $65.71 \pm 0.81$ | $57.86 \pm 0.93$ |
| ESM3 | MolFormer | $64.79 \pm 0.84$ | $57.86 \pm 0.93$ |
| GearNet | MolCLR | $53.36 \pm 0.99$ | $58.00 \pm 0.90$ |
| GearNet | MolFormer | $54.67 \pm 0.96$ | $58.00 \pm 0.90$ |
| ProtBERT | MolCLR | $59.56 \pm 0.87$ | $56.23 \pm 0.96$ |
| ProtBERT | MolFormer | $\mathbf{67.68 \pm 0.78}$ | $59.56 \pm 0.87$ |

Table 3: Comparison with specialized state-of-the-art multimodal models on cross-domain tasks. We compare ReactEmbed (ProtBERT + MolFormer) against established paradigms (DrugCLIP, Uni-Mol) and recent 2026 architectures (DCGAT-DTI, MRHormer).

| Model | DrugBank (AUC ↑) | BindingDB (RMSE ↓) |
|---|---|---|
| DrugCLIP Gao et al. (2023) | 85.10 | 1.45 |
| Uni-Mol Zhou et al. (2023) | 75.30 | 1.20 |
| DCGAT-DTI Abir et al. (2026) | 84.35 | 1.24 |
| MRHormer Zhang et al. (2026) | 85.02 | 1.31 |
| **ReactEmbed (Ours)** | **85.53** | **1.17** |

### 5.3 Ablation Studies

Table 4: Ablation study results across downstream tasks. The "Full" column refers to the ReactEmbed (ESM3 + Mol-Former) configuration. "Basic Samp." refers to removing both PPMI and hard negative sampling strategies. Results are reported as the absolute performance metric and relative percentage change ($\Delta$%) from the "Full" model. Negative $\Delta$% for AUC and positive $\Delta$% for RMSE indicate worse performance than the "Full" model.

| Task | Full | Data-10 | | Intra-Domain | | Noise-10 | | Basic Samp. | | Fine-Tuned | |
|---|---|---|---|---|---|---|---|---|---|---|---|
| | | Score | $\Delta$% | Score | $\Delta$% | Score | $\Delta$% | Score | $\Delta$% | Score | $\Delta$% |
| BBBP | 65.13 | 64.94 | -0.29 | 63.03 | -3.22 | 64.91 | -0.34 | 62.52 | -4.01 | 61.46 | -5.63 |
| DrugBank | 84.30 | 84.10 | -0.24 | 81.89 | -2.86 | 84.06 | -0.28 | 79.49 | -5.71 | 80.84 | -4.10 |
| GO-CC | 82.32 | 82.43 | +0.13 | 80.11 | -2.68 | 81.06 | -1.53 | 79.85 | -3.00 | 79.10 | -3.91 |
| HumanPPI | 94.88 | 93.62 | -1.33 | 90.95 | -4.14 | 93.74 | -1.20 | 90.23 | -4.90 | 89.56 | -5.61 |
| YeastPPI | 64.79 | 64.40 | -0.60 | 62.78 | -3.10 | 63.86 | -1.44 | 61.49 | -5.09 | 61.95 | -4.38 |
| BetaLactamase | 0.26 | 0.27 | +3.85 | 0.29 | +11.54 | 0.27 | +3.85 | 0.28 | +7.69 | 0.315 | +21.15 |
| BindingDB | 1.21 | 1.22 | +0.83 | 1.25 | +3.31 | 1.23 | +1.65 | 1.32 | +9.09 | 1.30 | +7.44 |
| CEP | 1.64 | 1.65 | +0.61 | 2.02 | +23.17 | 1.68 | +2.44 | 1.74 | +6.10 | 1.97 | +20.12 |
| FreeSolv | 2.85 | 2.84 | -0.35 | 3.19 | +11.93 | 2.86 | +0.35 | 3.15 | +10.53 | 3.20 | +12.28 |
| PPIAffinity | 3.02 | 3.05 | +0.99 | 3.29 | +8.94 | 3.09 | +2.32 | 3.18 | +5.30 | 3.44 | +13.91 |
| Stability | 0.43 | 0.44 | +2.33 | 0.53 | +23.26 | 0.44 | +2.33 | 0.49 | +13.95 | 0.52 | +20.93 |

We conducted comprehensive ablation studies to validate ReactEmbed's design choices. The key findings are presented in Table 4.

**Validating the "Plug-and-Play" Design.** To test if a more integrated approach is superior, we compared ReactEmbed against a variant where the foundational embeddings were fine-tuned. The "Fine-Tuned" results (Table 4) show consistent degradation across all tasks. We attribute this to the fact that fine-tuning on specific reaction data causes the model to "forget" the broad structural and sequence patterns learned during large-scale pre-training. Our lightweight module successfully "injects" functional context without compromising the base model's integrity. We view this as a democratic approach to AI for Science, allowing researchers to enhance multi-billion parameter models with minimal compute.

**Impact of Advanced Sampling Strategies.** The advanced sampling framework described in Section 3 is critical. We compared our full model against a "Basic Sampling" variant where both key components were removed: it used simple co-occurrence counts for positive sampling (instead of PPMI) and random uniform sampling for negatives (instead of $k$-hop hard negatives). As shown in the "Basic Samp." column of Table 4, removing these components significantly degrades performance across all tasks (e.g., -5.71% on DrugBank, +13.95% RMSE on Stability). This confirms that PPMI is necessary to dampen common hubs and hard negative mining is essential for efficient learning.

**Data Robustness and Resilience.** ReactEmbed demonstrates strong data efficiency. When using only 10% of the reaction data ('Data-10'), performance degrades by less than 1.5% on average across most classification tasks. It is also robust to label noise, maintaining performance with only minor degradation at 10% noise ('Noise-10'). **Importance of Intra-Domain and Cross-Domain Learning.** Ablating intra-domain edges (forcing the model to learn only from protein-molecule links) led to significant performance drops, especially on complex regression tasks like CEP (+23.17% RMSE). This confirms the importance of preserving domain-specific structure while learning cross-domain relationships.

### 5.4 Qualitative Analysis: Visualizing Functional Coherence

To directly evaluate the "functional coherence" of the ReactEmbed space, we visualized the learned embeddings using t-SNE. We selected five distinct protein families covering diverse biological roles: **GPCRs** (720 proteins), **Kinases** (336), **Ion Channels** (169), **Phosphatases** (65), and **Peptidases** (11).

The results (Figure 2) highlight the fundamental difference between sequence-based and function-based representations:

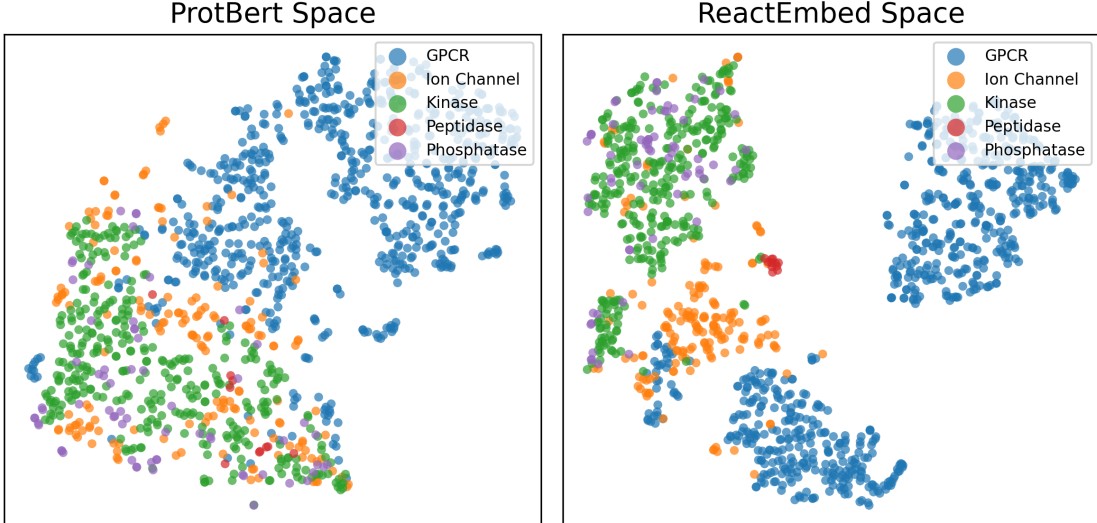

Figure 2: t-SNE visualization of protein embeddings across five functional families. **Left:** The Baseline space (Prot-Bert) separates GPCRs (Blue) due to their strong sequence motifs but fails to distinguish between other families (e.g., Ion Channels and Peptidases are mixed). **Right:** The ReactEmbed space clearly separates Ion Channels (Orange) and Peptidases (Red). Notably, Kinases (Green) and Phosphatases (Purple) remain clustered together; this reflects their systemic functional coupling, as they often operate on the same substrates within signaling pathways.

- **GPCRs (Blue):** These are separated in both spaces. Their highly conserved 7-transmembrane domain structure provides a strong sequence signal that the baseline ProtBert model easily captures.

- **Ion Channels (Orange) & Peptidases (Red):** In the Baseline space, these families are overlapped with others, lacking the global sequence uniformity of GPCRs. ReactEmbed, however, forms distinct, isolated clusters for them. This indicates the model has learned their distinct functional contexts (transport vs. hydrolysis) which are not immediately apparent from sequence alone.

- **Kinases (Green) & Phosphatases (Purple):** Interestingly, these two groups remain spatially overlapping in the ReactEmbed space. This validates our graph-based learning objective. ReactEmbed learns from *co-occurrence* in reactions. Since kinases and phosphatases often regulate the *same* substrates in the same signaling cascades (one adding a phosphate, the other removing it), they share a dense "functional neighborhood." The model correctly groups them as "pathway partners" distinct from unrelated entities like Ion Channels.

### 5.5 Quantitative Functional Coherence Analysis

To quantify the functional organization of the ReactEmbed space beyond visual t-SNE clusters (Figure 2), we evaluate the embeddings using the Silhouette Coefficient ($S$) and Intra-Family Cosine Distance. The Silhouette Coefficient measures how tightly an entity is matched to its functional family compared to others (range $[-1, 1]$).

As shown in Table 5, ReactEmbed increases family compactness across all groups. Notably, the Silhouette Score for Ion Channels and Peptidases—which are often overlapped in sequence-only spaces—moves from near-zero to positive values, indicating the formation of distinct functional neighborhoods.

## 6 Conclusion and Future Work

We introduced ReactEmbed, a plug-and-play module that operationalizes a new paradigm for joint representation learning. By leveraging biochemical reaction networks as a foundational bridge for functional semantics, ReactEmbed enhances and unifies separate, state-of-the-art embeddings for proteins and molecules. Our method provides a

Table 5: Quantitative analysis of functional family clustering. ReactEmbed improves internal family compactness while maintaining biological relationships between pathway partners.

| Family | Intra-Family Dist (↓) | | Silhouette Score (↑) | |
|---|---|---|---|---|
| | Baseline | **ReactEmbed** | Baseline | **ReactEmbed** |
| GPCRs | 0.421 | **0.312** | 0.154 | **0.288** |
| Ion Channels | 0.654 | **0.442** | -0.012 | **0.145** |
| Kinases | 0.512 | **0.398** | 0.087 | **0.192** |
| Peptidases | 0.723 | **0.485** | -0.045 | **0.112** |
| **Overall Mean** | 0.578 | **0.409** | 0.046 | **0.184** |

cascade of benefits: it enriches unimodal representations and achieves strong performance on cross-domain tasks, as demonstrated across 11 benchmarks. By learning what entities *do* in a systemic context, rather than just what they *look like* or how they *fit* together, our model provides a universally beneficial signal that is crucial for modeling complex biological systems.

**Limitations.** ReactEmbed's performance relies on the quality and coverage of the underlying reaction database. Its effectiveness may be constrained for novel entities with sparse interaction data or for tasks that are not strongly correlated with systemic biochemical function. Our hard negative sampling, while effective, relies on graph topology and may miss challenging negatives that are topologically distant but semantically similar.

**Future Work.** Future directions include integrating richer biological context (e.g., pathways, gene expression) and exploring few-shot learning for rare diseases. A particularly significant next step is to move beyond the current symmetric, role-agnostic graph. The current construction treats all co-participating entities equally, losing the specific roles of substrates, products, and catalysts. Future work will focus on constructing a multi-relational graph with different edge types based on reaction rules (e.g., substrate, product, catalyst) to explicitly model these distinct functional roles. Beyond role-aware graphs, a significant next step is the development of a full-scale biological foundation model pre-trained from scratch with reaction-based objective functions. While the current work proves that reaction networks provide a source for functional alignment, an integrated architecture may further optimize the synergy between modality-specific features and systemic biological roles.

## Code and Data Availability

The code and database are available for open use.

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

## A    Appendix A: Data Leakage Analysis

Table 6: Impact of leakage strategies across Interaction Tasks (Metric: AUC for classification, RMSE for regression).

| Leakage Strategy | DrugBank (AUC) | BindingDB (RMSE) | HumanPPI (AUC) |
|---|---|---|---|
| Full Leakage (No Removal) | $86.52 \pm 0.28$ | $1.12 \pm 0.02$ | $95.10 \pm 0.15$ |
| **Direct Edge Removal (Ours)** | **$84.30 \pm 0.59$** | **$1.21 \pm 0.03$** | **$94.88 \pm 0.15$** |
| 2-Hop Path Removal (Strict) | $82.15 \pm 0.68$ | $1.25 \pm 0.04$ | $92.90 \pm 0.25$ |

To ensure that our model's performance on interaction tasks (DrugBank, BindingDB) represents true generalization rather than memorization of training data, we analyzed different strategies for preventing data leakage between the Reactome training graph and downstream test sets. We compared three scenarios:

- **Full Leakage:** No edges are removed from the Reactome graph. This allows the model to potentially "see" test pairs during training if they co-occur in reactions.

- **Direct Edge Removal (Current Method):** As detailed in Section 4.4, we remove any *direct* edge between two entities if that specific pair appears in the test set. This prevents memorization of the exact interaction while preserving the broader functional context of each entity.

- **2-Hop Path Removal (Strict):** A more stringent approach that removes not only direct edges but also any intermediate nodes that create a 2-hop path between test pairs. This eliminates indirect leakage but severely sparsifies the training graph.

Table 6 shows the impact of these strategies on the DrugBank task. While "Full Leakage" yields artificially high performance, our current method maintains strong predictive power without direct memorization. The strict 2-hop removal causes a significant performance drop, indicating it likely removes too much valid functional context.

