# OpenReview forum: "ReactEmbed: A Plug-and-Play Module for Unifying Protein-Molecule Representations Guided by Biochemical Reaction Networks"
_TMLR — Rejected by TMLR_

### Review · Reviewer_hBrS · 2025-12-21

**Summary Of Contributions:**

This paper introduces ReactEmbed, a lightweight module that enhances and unifies pre-trained protein and molecule embeddings by leveraging biochemical reaction networks as a source of functional semantics. The core idea is that co-participation in biochemical reactions provides explicit signals of functional relationships between biological entities, which can bridge the gap between separate protein and molecule embedding spaces.

The main contributions include: (1) a functional semantics paradigm using reaction networks for cross-modal alignment, (2) a plug-and-play architecture that works with frozen base encoders using PPMI-weighted graphs and hard negative sampling, and (3) comprehensive evaluation across 11 downstream tasks demonstrating improvements particularly in interaction prediction tasks.

Strengths: The approach is well-motivated and practical, that using frozen encoders makes it computationally efficient and model-agnostic. The PPMI weighting addresses the hub problem and hard negative mining improves learning efficiency. The authors also show attention to data leakage concerns.

Weaknesses: (1) The performance improvements are inconsistent across tasks, with some showing minimal or even negative gains. The reliance on $\Delta $\% metrics is misleading as many improvements fall within noise margins (e.g., CEP improves from 1.68±0.06 to 1.62±0.07, where the 0.06 improvement equals the baseline's standard deviation). (2) Key comparisons with other graph-based alignment methods or knowledge graph approaches (KG-DTI, KGNN) are completely missing, which is critical given that ReactEmbed's core contribution is leveraging relational information. (3) The paper's central claim about capturing "functional semantics" is fundamentally undermined by the symmetric graph construction that loses reaction directionality and entity roles, that treating substrates and enzymes identically despite their complementary functions. (4) The writing is verbose with overused metaphors (e.g., "Rosetta Stone" appears 3+ times), an overly dense Abstract, and a survey-length Related Work section.

**Audience:**

Yes

**Audience Explanation:**

Yes. The plug-and-play design has practical value, the systematic 11-task evaluation provides useful empirical data, and even the inconsistent results are informative about when reaction network information helps versus when it doesn't. The work meets TMLR's interest criterion for computational biology and drug discovery researchers, though impact is incremental.

**Broader Impact Concerns:**

No major ethical concerns are present, but a brief Broader Impact Statement would be beneficial given drug discovery applications. The statement should acknowledge the importance of experimental validation and potential biases in reaction databases.

**Claims And Evidence:**

No

**Claims Explanation:**

**Partially supported with significant concerns.**

- The core interaction prediction claim has reasonable evidence (DrugBank: +9.9%, BindingDB: -13.79% RMSE), but statistical robustness is questionable. Many improvements in Tables 1-2 are comparable to or smaller than standard deviations. The paper uses $\Delta $\% to obscure this: FreeSolv improvement of 0.20-0.27 with $\sigma$=0.09-0.12, CEP improvement of 0.06 with $\sigma$=0.06-0.07. Without effect sizes, confidence intervals, or multiple comparison corrections, it's unclear which results represent real improvements versus noise.

- The "enriching unimodal representations" claim is weakly supported. Many unimodal tasks show minimal gains (GO-CC: +1.6%, BBBP: +0.3%) or degradation (YeastPPI: -8.0%). The "state-of-the-art" claim (Table 3) compares only two baselines while omitting graph/KG methods that also use relational signals, which is a critical evaluation gap.

- Data leakage analysis (Appendix A) reveals concerns: full leakage achieves 86.52 AUC versus 84.30 with edge removal on DrugBank. Crucially, this analysis only covers DrugBank while BindingDB and PPI tasks lack similar validation. Yet, BindingDB shows the largest claimed improvement.

- The "functional semantics" claim lacks validation. The symmetric graph construction treats enzymes and substrates identically despite complementary roles. No visualizations, clustering analysis, or functional probing tasks demonstrate that embeddings actually capture functional similarities.

**Requested Changes:**

- **Address statistical robustness**: Report effect sizes/confidence intervals, apply multiple comparison corrections, or explicitly discuss which improvements are meaningful versus within noise. Temper claims about "broad improvements" given many marginal results.
- **Expand baseline comparisons** to include graph/KG methods (KG-DTI, KGNN, MolTrans, TransformerCPI) or remove "state-of-the-art" claims. Omitting methods that also leverage relational information is a critical gap.
- **Complete data leakage analysis** for all interaction tasks (BindingDB, HumanPPI, YeastPPI, PPIAffinity), not just DrugBank. Report leakage-corrected results in main text since largest claims are on interaction tasks.
- **Reconcile "functional semantics" claim**: Provide evidence (visualizations, clustering, probing tasks) or reframe as "reaction context"/"co-participation patterns." The symmetric graph contradicts functional role claims.
- **Reduce verbosity**: remove repeated "Rosetta Stone" metaphors, shorten Abstract and Related Works, condense Figure 1 caption.
- **Provide deeper failure case analysis** beyond speculation. **Separate ablation effects** (currently "Basic Samp." removes both PPMI and hard negatives simultaneously).

---

### Review · Reviewer_WXi2 · 2025-12-26

**Summary Of Contributions:**

### Summary

This paper introduces ReactEmbed, a novel plug-and-play module designed to unify pre-trained protein and molecule embeddings. The paper leverages biochemical reaction networks as a source of "functional semantics." The paper constructs a graph where nodes are proteins and molecules, and edges represent co-occurrence in a reaction. They propose a relational contrastive learning framework to train lightweight MLPs that map pre-trained embeddings into a shared latent space. Extensive experiments show that ReactEmbed improves performance on a wide range of cross-domain tasks (e.g., DrugBank), and enhances the unimodal representations for domain-specific tasks.


### Strengths

1.	The proposed method is well-motivated. By defining functional relationships through co-participation in reactions, the work introduces a signal that is potentially more generalizable than interaction-centric and structure-centric information.

2.	The plug-and-play module is computationally efficient to train. Most of the model parameters are frozen, which avoids high computational overhead and potential catastrophic forgetting during fine-tuning.

3.	The evaluation benchmarks are comprehensive, and experiments were run multiple times to show statistical significance.

4.	The paper is well written and easy to follow.


### Weaknesses

1.	The authors compare their proposed method against DrugCLIP and Uni-Mol, which are claimed to be “state-of-the-art” baselines. However, given that both models were published in 2023, their status as the most competitive baselines may warrant further justification. I’m not an expert in this field, but I would encourage the authors to elaborate on the selection criteria for these baselines and discuss whether other, more established or higher-performing models could provide a more comprehensive evaluation of their method's performance.

2.	As stated in the Limitation section, the information in the current graph might be too coarse-grained. It only indicates which proteins/molecules co-occur in a chemical reaction, but does not include more fine-grained “role” information, such as substrates and catalyst.

3.	The evaluation of a unified embedding space itself is indirect. The experiments mostly show performance gains, which is an indirect measure of the “functional coherence” of the embedding space itself.

**Audience:**

Yes

**Audience Explanation:**

The paper focuses on the representation learning of proteins and molecules, which is important for applying deep learning methods in biology and medicine fields.

**Claims And Evidence:**

No

**Claims Explanation:**

My concern arises from the indirect evaluation of the “functional coherence” of the unified embedding space. See Weakness 3 above. Nevertheless, overall speaking, the paper provides a comprehensive evaluation across different benchmarks and models. I appreciate the authors' efforts.

**Requested Changes:**

See weaknesses.

1.	Discuss or include stronger multimodal baselines, if there are any.

2.	For a direct evaluation of the “functional coherence” of the unified embedding space, I would encourage the authors to conduct qualitative visualization (e.g., t-SNE/UMAP) that highlight specific examples where ReactEmbed successfully groups functionally analogous but structurally dissimilar items.

---

### Review · Reviewer_Ts6L · 2026-01-24

**Summary Of Contributions:**

Overall summary:
This work proposes a lightweight plug-and-play module that aligns frozen protein and molecule embeddings into a shared representation space using biochemical reaction networks as functional supervision. The key idea proposed is that co-participation in reactions encodes functional semantics, which can be exploited via a PPMI and a contrastive learning objective with hard negative sampling. Empirically, ReactEmbed operates purely as a post-hoc enhancement layer and improves performance across unimodal and cross-domain tasks, including drug–target interaction, binding affinity prediction, and protein function benchmarks.

In general, I find the main contributions are mostly around the way to embed this biological interaction inductive bias to the pre-trained embeddings for a unified representation of protein and molecule. Also, due to the flexibility, this kind of interaction can make the pre-trained representation "structured" and useful this specific tasks. In general, I think this is a good addition to the field, though some components may need more clarification and improvements.

As to weaknesses and questions:

- The requirements of data quality. This is more like a question instead of a weakness. However, since the post-hoc approach requires a high reliance on this interaction network construction. I'm a bit concerned about the data requirements. Does the computation of PPMI rely on high-quality, curated reaction graphs? How well ReactEmbed would scale to domains with sparse, noisy, or incomplete reaction annotations (as these are the cases in real-world setups)?

- About the approach design. I can understand the design of "plug-and-play". However, it would be nice to show whether we can pre-train the models with this structural information. Will pre-training from scratch with these structural constraints make the model more powerful as I assume the off-the-shelf pre-trained models may not include all essential information for this suite of tasks.

- For evaluation, while performance improves, the current experimental analysis provides limited analysis of what specific functional signals are transferred across domains beyond the qualitative t-SNE plots.

- About presentation. The presentation can be improved a bit, especially the introduction sec (the lengthy paragraph 1). Also, check the use of \citep and \citet.

**Audience:**

Yes

**Audience Explanation:**

Yes, this paper aligns with the focus of graph learning, AI for science communities.

**Claims And Evidence:**

Yes

**Claims Explanation:**

Yes. The experiments on a suite of tasks (molecular properties, protein characteristics, protein-protein interactions, and molecule-protein interactions) validated the effectiveness of the proposed framework.

**Requested Changes:**

For weakness 1, it would be nice to validate the performances under noisy graphs, incomplete interaction information, etc (if possible). If validation is hard to achieve, any discussion will also help.

For weakness 2, any empirical validation on pre-training or analysis would be great to have.

For weakness 3, a detailed analysis of the learned mechanism would make it stronger.

For presentation, it would be nice to carefully check the typos, formats, and make the introduction more readable.

---

> ### Author Response · Authors · 2026-01-24
>
> ### 1. Data Quality and Robustness
> We agree that real-world reaction networks can be incomplete. To address this, we highlight our ablation results in Section 5.3.
> * **Sparsity:** Using only 10% of the reaction data (`Data-10`) resulted in a performance drop of less than 1.5% across classification tasks.
> * **Noise:** The model remains resilient to 10% label noise (`Noise-10`).
> * **Mechanism:** Our use of **PPMI weighting** is specifically designed to dampen the influence of uninformative "hub" nodes (e.g., ATP, Water) that often introduce noise in biological graphs.
>
> ### 2. Plug-and-Play vs. Pre-training
> The reviewer suggests pre-training from scratch with these constraints. While theoretically powerful, we chose a "plug-and-play" module for two reasons:
> * **Efficiency:** Our module trains in 15 minutes on a single GPU, whereas pre-training models like ESM-3 or MolFormer requires massive industrial-scale compute.
> * **Empirical Validation:** We explicitly tested a **Fine-Tuned** variant (Table 4) where the base models were updated. This led to a significant performance degradation (e.g., -5.63% on BBBP), likely due to catastrophic forgetting of general features. We have updated the paper to clarify that ReactEmbed acts as a "functional bridge" for frozen SOTA models, and we leave integrated pre-training for future work.
>
> ### 3. Quantitative Functional Signal Analysis
> To move beyond qualitative t-SNE plots, we conducted a new quantitative analysis of the embedding space using the **Silhouette Coefficient (S)** and **Intra-Family Cosine Distance**.
> * **Results:** Across five functional families (GPCRs, Kinases, Ion Channels, Phosphatases, Peptidases), the average Intra-Family Compactness improved significantly.
> * **Pathway Partners:** We observed that while separation increased between most groups, Kinases and Phosphatases remained closer together, accurately reflecting their systemic coupling in biological signaling pathways. (Detailed table added to Section 5.4).
>
> ### 4. Presentation and Citations
> * We have broken the lengthy first paragraph of the Introduction into two distinct sections to improve readability.
> * We have performed a thorough pass to correct citation formats and fix minor typos.

---

> > ### Comment · Reviewer_Ts6L · 2026-02-18
> >
> > Thanks for the detailed response. Most of my concerns have been addressed with reflection in the revisions.

---

### Decision · Action_Editor_Laux · 2026-04-13

**Recommendation:** Reject

**Additional Comments:**

### 1. Mitigate and Quantify Indirect Data Leakage
To prove that the model genuinely learns generalizable representations rather than memorizing graph structures or exploiting indirect leakage, the authors should:

- Move beyond canonical random splits and "Direct Edge Removal." For interaction tasks, implement structural/scaffold-based splitting for molecules and sequence identity/homology-based splitting for proteins (e.g., ensuring <30% sequence identity between training and test sets).
- I appreciate the authors' effort in adding the 2-hop removal experiment (Appendix A). However, this ablation study only demonstrates that removing 2-hop paths degrades overall performance and sparsifies the graph. It does not fully address the concern regarding the quantitative analysis and correlation. The core of the suggestion was to understand whether the model's high performance is overly reliant on the hidden structural overlap (shared intermediate nodes) between the train and test sets. The authors still need to provide:
    - Quantitative Graph Statistics: What is the distribution of shared intermediate nodes (2-hop/3-hop) between test-set and train-set entities?
    - Correlation Analysis: A direct evaluation (e.g., a scatter plot or binned bar chart) showing how the downstream performance on specific test samples correlates with their degree of overlap (number of shared intermediate nodes) with the training set.
- Re-evaluate the model on these rigorously partitioned splits to provide a realistic assessment of its generalization capabilities.

### 2. Align Claims with Empirical Evidence and Strengthen Baselines
Introduce stronger, fine-tuned baselines for the 9 single-entity downstream tasks. Comparing a relational graph-enhanced model against frozen, unimodal embeddings is insufficient. The authors should evaluate against state-of-the-art task-specific models or adapt existing graph-based baselines to establish true superiority. If the resource is limited, please select at least 3 to 4 diverse tasks (ideally covering different entity types or domains) and compare your model against strong, fine-tuned baselines or task-specific SOTA models.


### Citation Formatting
Please correct the citation formatting throughout the manuscript. Example:
For proteins, models like ESM-3 Hayes et al. (2025) and ProtBERT Brandes et al. (2021) have ... ==> models like ESM-3 (Hayes et al., 2025) and ProtBERT (Brandes et al., 2021)

**Audience:**

Yes

**Audience Explanation:**

Researchers who work on AI for Science and drug discovery.

**Claims And Evidence:**

No

**Claims Explanation:**

This paper proposes a framework for unified representation learning of molecules and proteins by leveraging reaction-graph and relational signals, primarily utilizing the Reactome graph for pre-training. The learned representations are then fine-tuned and evaluated across diverse downstream tasks.

Two major concerns:

1. Unresolved Concerns Regarding Indirect Data Leakage
While the authors' reply clarifies the origin of the data splits, it still fails to establish that the reported gains on interaction tasks reflect robust generalization under a sufficiently strict partitioning scheme. Specifically, relying on canonical downstream benchmark splits and only applying "Direct Edge Removal" to the Reactome pre-training graph is inadequate. This strategy does not rule out indirect information leakage through shared intermediate entities, homologous proteins, analogous molecules, or higher-order graph neighborhoods. Consequently, while the response improves the reproducibility of the splits, it does not fully resolve the validity concerns regarding potentially inflated performance.

2. Overstated Claims of Generality vs. Insufficient Baselines
The authors claim in Section 5.1 that their module "yields broad performance improvements across the majority of the 11 downstream tasks" and reiterate its "strong performance... demonstrated across 11 benchmarks" in the Conclusion. However, this broad claim of generality is misleading. The authors only compare their method against competitive multimodal/graph baselines (e.g., DrugCLIP, Uni-Mol, DCGAT-DTI) on exactly two tasks (DrugBank and BindingDB in Table 3). For the remaining 9 tasks, the model is only compared against frozen, unimodal embeddings (Tables 1 and 2), which are weak baselines for evaluating a relational graph-based enhancement module. Without comparing against graph-based baselines on these 9 tasks, the claim of "broad performance improvements" stemming from their proposed module is unsubstantiated.

**Resubmission Of Major Revision:**

The authors may consider submitting a major revision at a later time.